# Correlations between Gut Microbial Composition, Pathophysiological and Surgical Aspects in Endometriosis: A Review of the Literature

**DOI:** 10.3390/medicina59020347

**Published:** 2023-02-12

**Authors:** Irene Iavarone, Pier Francesco Greco, Marco La Verde, Maddalena Morlando, Marco Torella, Pasquale de Franciscis, Carlo Ronsini

**Affiliations:** 1Department of Woman, Child and General and Specialized Surgery, Obstetrics and Gynecology Unit, University of Campania “Luigi Vanvitelli”, Largo Madonna delle Grazie 1, 80138 Naples, Italy; 2Department of Woman, Child and Public Health, A. Gemelli, IRCCS, University Hospital Foundation, 00168 Rome, Italy

**Keywords:** endometriosis, estrogen, gut microbiota, microbiome, abdominal hysterectomy, inflammation

## Abstract

*Background and Objectives*: Endometriosis is an estrogen-dependent, inflammatory, gynecological disorder represented by the migration of endometrial tissue outside the uterus. It can manifest through gynecological and gastrointestinal (GI) signs. Given the hormonal imbalances in endometriosis and the effect of microbiota on immune dysfunction, it has been thought that the human microbiome may play a role in its pathogenesis, acting differently before and after laparotomy. The aim of this review is to establish whether there is an interaction between endometriosis and gut microbial composition. *Materials and Methods*: We aimed to review available literature by systematically searching five databases: PubMed, EMBASE, Scopus, Cochrane Library, and ScienceDirect. We included records describing gut microbiota in the context of endometriosis—observing PRISMA (Preferred Reporting Items for Systematic Reviews and Meta-Analysis) guidelines—to recognize the presence of disease by the expression of bacterial taxa—based on 16S ribosomal RNA gene sequencing analysis. *Results*: Among 10 studies selected, there were four review articles and six clinical trials. The latter identified significant differences at a genus level in increased Prevotella, Blautia, and Bifidobacterium and decreased Paraprevotella, Ruminococcus, and Lachnospira (*p* < 0.05). In patients undergoing abdominal hysterectomy, Proteobacteria phylum increased from 34.36% before surgery to 54.04% after surgery (*p* < 0.05). *Conclusions*: Although scientific literature reports different characterizations of intestinal microbiota in endometriotic patients, further evidence is needed to develop new diagnostic-therapeutic strategies, for example, administration with probiotics before surgery.

## 1. Introduction

Estrogen levels in postmenopausal patients are associated with both richness of species in gut microbiota and levels of Clostridia taxa in fecal samples [1]. Given the hormonal dysfunction in patients with endometriosis and the impact of microbiota on immune imbalances, it has been hypothesized that the gut microbiome may play a role in the pathogenesis of endometriosis [2]. It would be appropriate to identify a core microbiota in endometriotic patients. The aim of this review was to establish whether there is an interaction between endometriosis and gut microbial composition.

There is convincing evidence of an interaction between the neuroendocrine, immune system, and gut coexisting in gynecological conditions like endometriosis [3]. Alterations in gut microbial composition, known as dysbiosis, influence immunosurveillance [4]. The uterine cavity hosts a far-less investigated microbial community, which concerns the female reproductive tract (FRT) entirely, although there is no consistent approval on the key FRT microbiota; hence, there is no supporting evidence about its function in endometriosis [4]. Immune response results in imbalances in cell expression patterns and increasing levels of proinflammatory cytokines. Endometriosis progression, on the other hand, may parallel the persistence of these conditions, setting into a chronic state of inflammation with cell adhesion and angiogenesis over time [3]. That is set up by interleukin-6 (IL-6), IL-8, Vascular Endothelial Growth Factor (VEGF), and suppression of cell-mediated immunity. They regulate both primary establishments and the persistence of the disease [4]. Antibiotics and probiotics show the ability to reduce symptoms of endometriosis, such as abdominal pain [3]. Gut bacteria regulate proteolysis reactions: for example, serotonin is synthesized from tryptophan amino acid; dopamine, noradrenaline, and adrenaline are derived from tyrosine [5]. The ability of those essential amino acids to enter the brain gateway depends on their plasma concentrations: decarboxylation of tryptophan promotes tyrosine synthesis. The uptake of tryptophan originates the so-called Acute Phase Reaction (APR), generating stress [6,7]. In dysbiosis, gut flora is altered in its enzymatic activity, becoming unable to provide the Central Nervous System with neurotransmitters [5], whereas a eubiotic ecosystem is represented by intense vitamin biosynthesis like cobalamin, and catabolism of xenobiotics, which could be involved in inflammation [5]. The 50% increased risk of inflammatory bowel disease (IBD) in endometriosis-affected patients demonstrates a solid link between immunological responses into the gut and endometriotic injuries [3]. The recognition of an endometriotic signature in intestinal microbiota would pave the way for therapeutic approaches with probiotics and prebiotics before considering surgery. One treatment for endometriosis is abdominal hysterectomy—in order to remove endometrium and uterus—and possibly bilateral salpingo-oophorectomy [6,8]. There are extremely little data regarding surgery influencing intestinal flora. The expected results of our study are strongly influenced by the small number of existing works.

## 2. Materials and Methods

### 2.1. Search Method

The methods for this study were specified a priori based on the recommendations in the preferred reporting items for systematic reviews and meta-analyses (PRISMA) statement [9].

We systematically explored 5 search engines to establish eligible studies: PubMed, EMBASE, Scopus, Cochrane Library, and ScienceDirect in March 2022. No restriction for country and year of publication was performed. We adopted the following string of idioms to identify studies that were fitting to the topic of our review: “Endometriosis and microbiota.” We collected the accessible literature in the English language, focusing on the characteristics of gut microbial composition compared to clinical and pathophysiological aspects of endometriosis. Two examiners (I.I. and P.F.G.) separately screened the literature. In case of discrepancy, C.R. decided on inclusion or exclusion. Eligible articles were selected in three phases: at first, records that were unsuitable to the topic of our review were removed. Second, we deleted duplicates (i.e., the same record shown in multiple datasets) and articles by selection from the abstract. In the third screening time, studies were selected after full-text analysis. This review embraced review articles and prospective trials describing the gut microbial composition of humans in the context of endometriosis to identify a specific pattern of disease defined by the expression of bacterial taxa. Exclusion criteria included articles involving animal subjects, non-original studies, pre-clinical trials, abstract-only publications, and articles in a language other than English. The studies selected and the reasons for exclusion are mentioned in the PRISMA flow diagram (Figure 1). The present review is categorized on PROSPERO (ID: CRD42022354457).

### 2.2. Microbial Analysis

To standardize the studies’ results, we enrolled all the studies that used the 16S ribosomal RNA gene sequencing-based method to identify bacterial taxa in stool samples. Beta-diversity between samples on both weighted and unweighted UniFrac distance metrics was performed to compare dissimilarity through QIIME2 [10]. An ANOVA Repeated Measures was used as a variance test when investigating alpha-diversity [11].

## 3. Results

In the primary research, 240 records from 2016 to 2021 were obtained from PubMed, EMBASE, Scopus, Cochrane Library, and ScienceDirect search engines. According to the PRISMA flow diagram, we deleted 198 articles that were unsuitable to the topic of our review. Among the 42 records remaining, we excluded 15 duplicates, and subsequently, among the 27 abstracts remaining, we removed 14 unfitting records by abstract selection. The number of full-text studies assessed for eligibility was 13, including one article with no access to data, one article in Chinese, and two abstract-only publications were excluded. Overall, nine records were finally chosen for analysis (Figure 1). The countries where the studies were conducted, publication years, study designs, and the number of participants are summarized in Table 1.

A total of 396 patients were included in our review. Among the 9 studies selected, there were 4 3 review articles and six clinical trials, paralleling pathophysiological features in patients with endometriosis to their gut microbial composition, compared to healthy controls. The main results of the studies are summarized in Table 2.

The differences in microbial genera expression profiles are summarized as follows.

### 3.1. Over-Expressed Microbial Taxa in Endometriosis

According to beta-diversity analyses with weighted and unweighted UniFrac distance metrics, Le et al. showed that the most prevalent phyla in fecal samples of patients with endometriosis compared to healthy controls, both with and without oral contraceptive pills (OCPs) administration were Firmicutes and Bacteroidetes, in two different scenarios: at day of surgery (DOS) and post-surgical intervention (PSI). At DOS, patients without OCPs administration showed 55.7% of Firmicutes and 33.7% of Bacteroidetes; at PSI, they resulted in 50.6% of Firmicutes and 42.4% of Bacteroidetes. Vice-versa, healthy controls showed 44.7% of Firmicutes and 40.1% of Bacteroidetes at DOS, ending in 52.8% of Firmicutes and 35.9% of Bacteroidetes at PSI. With OCP administration, patients had a gut composition of 43.8% Firmicutes and 45.2% Bacteroidetes at DOS, resulting in 52.7% of Firmicutes and 40.2% of Bacteroidetes at PSI; controls showed 41.8% of Firmicutes and 51.7% at DOS, ending in 45.3% of Firmicutes and 35.9% of Bacteroidetes at PSI. At DOS, GI bacteria were similar between endometriotic patients and healthy controls without OCPs administration but differed significantly with OCPs (unweighted *p* = 0.001, weighted *p* = 0.029). At PSI, GI bacteria of OCPs-administered patients were more similar to the controls’ (unweighted *p* = 0.165, weighted *p* = 0.424) [18]. As shown in Table 2, the gut microbiota of OCPs-and-surgery-administered patients showed a prevalence of Bacteroides and shortly after Prevotella, Blautia, Faecalibacterium, Dialister, Coprococcus, and Sutterella. Surgery with no association with medical treatment contributed to the abundance of Blautia and Dialister and reduced all other genera [18]. So long, endometriotic patients had a higher Firmicutes/Bacteroidetes ratio—1.2—compared to healthy controls—1 [18]. A finding by Ata et al. was that patients undergoing sigma and/or rectum resection due to endometriosis infiltrating nodules had a higher abundance of Enterobacteriaceae Escherichia/Shigella (82% and 7% in each patient; *p* < 0.01) [14]. In their study, Svensson et al. showed a significantly higher alpha-diversity (*p* = 4.9 × 10^−5^) in 198 healthy controls rather than in 66 endometriosis patients. They valued as median and interquartile range or number and percentage and demonstrated in endometriosis patients a more elevated level of Bacteroides (16.08 in endometriosis vs. 15.29 in controls; *p* < 0.05) and Parabacteroides (11.92 in patients vs. 11.27 in controls; *p* < 0.05)—belonging to the class of Bacteroidia— Oscillospira (11.79 in patients vs. 10.67 in controls; *p* < 0.05) and Coprococccus (10.81 in patients vs. 10.31 in controls; *p* < 0.05)—belonging to the class of Clostridia—and one unmarked genus of Gammaproteobacteria (4.38 in patients vs. 3.28 in controls; *p* < 0.05) [16]. After stratification by GI involvement, the authors noticed that women with GI nodules from endometriosis showed a higher abundance of Lactococcus—Bacilli class—compared to patients without GI involvement (3.90 vs. 2.30, respectively; *p* < 0.05), as well, they presented a higher abundance in Prevotella (0.00 vs. 4.96—*p* < 0.05), belonging to Bacteroidia class [16]. Moreover, the authors performed an analysis of bacteria in patients with symptoms and their resolution. Prevotella was associated with constipation (R = 0.307, *p* = 0.014), bloating and flatulence (R = 0.297, *p* = 0.016), and vomiting and nausea (R = 0.295, *p* = 0.017) [16]. At the phylum level, Firmicutes/Bacteroidetes ratio in patients with endometriosis was increased compared to controls (3.55 vs. 1.99, respectively; *p* < 0.05) [19]. Moreover, in endometriosis, the abundance of Actinobacteria, Cyanobacteria, Saccharibacteria, Fusobacteria, and Acidobacteria was significantly increased. Indeed, Actinobacteria were 13% in endometriosis patients and 4% in the control group, and Acidobacteria were completely absent in controls (*p* < 0.05) [19]. A genus analysis proved to significantly increase the abundance of Bifidobacterium, Blautia, Dorea, and Streptococcus. Prevotella was identified only in patients with endometriosis (78.8%), whereas Coprococcus was found just in healthy controls (65.15%) [19]. Regarding proinflammatory cytokines, as recently demonstrated [19], serum levels of IL-17A were positively correlated with Bacteroides abundance (R = 0.89, *p* < 0.05). Huang et al. further noticed that the most abundant phyla in stool samples of endometriosis patients were Firmicutes, Bacteroidetes, and Proteobacteria (>90% prevalence for all) [17]. Clostridiales Ruminococcaceae was the most abundant taxon in stool samples (average relative abundance: 0.21 ± 0.09 in controls, 0.17 ± 0.11 in endometriosis—expressed as mean ± Standard Deviation; *p* < 0.05) [17]. Moreover, Bacteroidaceae Bacteroides, Lachnospiraceae Blautia, Clostridiales Lachnospiraceae, Faecalibacterium prausnitzii, and Bifidobacteriaceae Bifidobacterium were the other most prevalent taxa (*p* < 0.05) [17].

### 3.2. Under-Expressed Microbial Taxa in Endometriosis

In endometriosis patients with only-OCP administration, Le et al. detected a lower abundance of Faecalibacterium. Bacteroides were prevalent in OCPs administrated controls (10%) but lower in endometriosis patients treated with OCPs at DOS (4%) [18]. At DOS, GI bacteria differed significantly with OCPs (unweighted *p* = 0.001, weighted *p* = 0.029) [18]. According to Jiang et al., endometriotic microbiota is represented by a lower presence of Lactobacillus [4]. In Ata et al. clinical trial, in stool samples of endometriotic patients, with stages 3 and 4 of the disease, genera Sneathia (*p* = 0.001), Barnesella (*p* = 0.001), and Gardnerella (*p* < 0.01) were significantly decreased compared to the healthy controls [14]. In particular, 13 patients over 14 showed complete absence of Sneathia, whereas only in 4 controls over 14 Sneathia was completely absent [14]. Concerning Barnesella, patients showed a relative abundance between 0.00 and 0.01 Operational Taxonomic Units (OTUs), whereas healthy controls’ relative abundance ranged between 0.00 and 0.05 OTUs [14]. Svensson et al. expressed values as median and interquartile range or number and percentage: Paraprevotella in Bacteroidia class (0.00 in endometriotic patients vs. 0.71 in healthy controls), Lachnospira in the Clostridia class (3.47 in patients vs. 12.43 in controls), Turicibacter in the Bacilli genus (2.89 in patients vs. 4.50 in controls) and one unmarked bacterium of the Coriobacteriia class (6.95 in patients vs. 8.24 in controls) were reduced (*p* < 0.05 for all) [16]. Compared to healthy controls, patients with GI symptoms had a lower abundance in SMB53—Clostridia class—(4.72 vs. 6.96, respectively; *p* = 0.01) and in Odoribacter (0.00 vs. 3.06, respectively; *p* = 0.02), belonging to Bacteroidia class [16]. Moreover, in endometriosis abundance of Tenericutes was significantly decreased (0% in patients vs. 2% in controls; *p* < 0.05) [19]. A genus analysis proved that Lachnospira (Eubacterium) was significantly decreased (1% in patients vs. 4% in controls; *p* = 0.00007), and whereas Prevotella was identified only in patients with endometriosis (78.8%), Coprococcus was found just in healthy controls (65.15%) [19]. Regarding proinflammatory cytokines, serum levels IL-17A were negatively correlated with Streptococcus (R = −0.89, *p* < 0.05) and Bifidobacterium (R = −0.89, *p* < 0.05) abundance. Moreover, as shown by Shan et al., serum IL-8 levels were negatively correlated with the Subdoligranulum abundance (R = −0.95, *p* < 0.05) [19]. Huang et al. further noticed that stool samples of endometriosis patients had significantly lower microbial richness according to the Shannon index (Mann–Whitney U test, *p* = 0.013) [17]. Beta-diversity (at principal coordinates analysis—PCoA) and Shannon index were significantly different in stool samples of endometriotic patients compared to healthy controls (2.66 ± 0.21, 2.38 ± 0.35, respectively, expressed as mean ± Standard Deviation) [17]. In patients, compared to controls, average relative abundances of Clostridia Clostridiales (0.01 vs. 0.25), Lachnospiraceae Ruminococcus (0.005 vs. 0.015), Clostridiales Lachnospiraceae (0.01 vs. 0.015), and Ruminococcaceae Ruminococcus (0.002 vs. 0.01) were decreased (*p* < 0.05) [17]. Xu et al. had already noticed that average taxa abundances in endometriosis patients, compared to controls, of Paraprevotella (0 vs. 125), Odoribacter (13 vs. 30), Veillonella (0 vs. 3), and Ruminococcus (300 vs. 1000) were significantly decreased in chronically stressed patients affected by endometriosis [13].

### 3.3. Gut Microbiota of Patients with Undergoing Abdominal Hysterectomy

There are few findings regarding gut microbial modifications among patients who underwent laparotomy. Abdominal hysterectomy influences intestinal flora, decreasing abundance and Shannon alpha-diversity of microorganisms. In particular, 16S ribosomal RNA gene sequencing revealed a higher abundance of Proteobacteria in patients with a history of abdominal hysterectomy [20]. That may be due to a reduction in estrogen levels. There is evidence that estrogen decrease is not significant in women administered with various ranges of hysterectomy, even though hormones are significantly reduced in patients administered with total hysterectomy [21]. At the phylum level, the dominant strains in the intestinal community both before and after abdominal hysterectomy were: Bacteroidetes, Proteobacteria, and Firmicutes, with a relative abundance of 75%. Although, before hysterectomy, the relative abundance of Bacteroidetes was higher (24.54%, *p* = 0.003), whereas, after hysterectomy, Bacteroidetes were reduced (11.43%, *p* = 0.003), and Proteobacteria were significantly increased in their relative abundance (from 34.36% to 54.04%, *p* = 0.016) [20]. Firmicutes did not differ significantly before and after surgery, but their proportions increased from 0.003% to 17.26% (*p* = 0.926) [20]. A T-test determined whether differences in bacterial communities were statistically significant at each taxonomic ranking (phylum, class, order); at the phylum analysis, Proteobacteria increased from 34.36% before surgery to 54.04% after surgery (*p* < 0.05); at a class level, Gammaproteobacteria (Proteobacteria phylum) increased from 22.74% before hysterectomy to 48.89% after hysterectomy (*p* < 0.05); at the order level, Enterobacteriales (Proteobacteria phylum, Gammaproteobacteria class) increased from 9.44% before surgery to 42.05% after surgery (*p* < 0.05). Significant differences were also found for the Verrucomicrobia phylum, which changed its relative abundance from 0.04% before hysterectomy to 0.13% after hysterectomy (*p* < 0.05) [20]. Unfortunately, there is a lack of evidence concerning gut microbiota, disease regression, and follow-up in patients treated with laparotomy.

## 4. Discussion

A healthy microbiome is characterized by a richness of species, identified by microbial diversity parameters such as the Shannon index of alpha diversity [22]. The intestinal microbiota hosts a fungal and viral community also. Most viruses called *phages* are useful for the bacterial community, transmitting antibiotic-resistance genes from one family to another, confirming that a diverse microbial community maintains homeostasis [23]. Sokol et al. identified the ratio between two fungal phyla, Basidiomycota and Ascomycota, showing inverse correlation to each other in with flare, IBD in remission, and healthy controls [24]. On the other hand, intestinal dysbiosis is mostly defined by bacteria, particularly Firmicutes and Bacteroidetes, representing the Gram-positive and Gram-negative populations [25]. Firmicutes/Bacteroidetes ratio is increased in IBS patients differing from healthy controls in their microbial composition [25]. Regarding Firmicutes/Bacteroidetes ratio, it is a focal point for dysbiotic gut flora. On the other hand, it is important to analyze the genus level also since Firmicutes include extremely eubiotic bacteria such as Lactobacilli, as well as dysbiotic bacteria like Clostridium difficile. The only endometriotic phenotype evincible from scientific literature [13,14,16,17,18,19] describes increased Prevotella, Blautia, and Bifidobacterium genera and decreased Paraprevotella, Ruminococcus, Lachnospira (*p* < 0.05). In addition, Svensson et al. recognized a distinctive biomarker, the Prevotella genus, in a correlation between bacteria and endometriosis symptoms resolution [16]. Shan et al. recognize Prevotella as significantly increased and identify significant differences in Blautia, Dorea, and Streptococcus [19]. Otherwise, an abundance of other genera, such as Cyanobacteria, Saccharibacteria, Fusobacteria, Bifidobacterium, Blautia, Dorea, and Streptococcus, is not clearly defined. The same bias was found for Bacteroides, Blautia, Clostridiales, Faecalibacterium prausnitzii, and Bifidobacterium in Huang et al. study and exclusively for Faecalibacterium in Le et al. study [17,18]. In Le et al. clinical trial, it would have been useful to distinguish surgical procedures between laparoscopic or laparotomic hysterectomy in order to assess gut microbial modifications in endometriotic patients before and after their surgical intervention [18]. In their literature review, Jiang et al. confirmed that the so-called “endometriotic microbiota” generally consists of a lower abundance of Lactobacilli and a greater abundance of opportunistic and vaginosis-related species, both in the gut and female reproductive tract, but they did not estimate prevalence and significance of this data [4]. Anyway, those data may suggest a healthier gut flora in the control group compared to women affected by endometriosis. Previously, Xu et al. identified the estrogen-gut-brain axis, hypothesizing that patients with endometriosis also suffer from chronic stress through the activation of β-adrenergic signaling in a vicious cycle [13]. In pre-clinical trials, murine models of endometriosis show increased Firmicutes/Bacteroidetes ratio and Bifidobacterium [26]. In human studies also, Firmicutes exceeded the Bacteroidetes phylum, but their ratio was not significantly increased [18]. At the same time, it is reported that metronidazole can reduce the inflammatory component of the lesions, lowering the Bacteroidetes phylum in mice (*p* < 0.01) [27]. Moreover, Enterobacteriaceae Escherichia/Shigella are among the prevalent dysbiotic taxa in IBD and Irritable Bowel Syndrome (IBS) [2]. Endometriosis is also linked to lower Lactobacillus abundance and the prevalence of gut inflammation in rhesus monkeys [28]. The lower microbial richness and beta diversity according to the Shannon index in patients with endometriosis compared to healthy controls demonstrate that multiple bacteria may contribute to variability in gut microbial composition [17]. Dysbiosis is a dysfunctional impairment of the microbiota. It can derive from multiple augmented pathogens or loss of pre-/probiotics, with consequences on IBD also [29]. Besides some discrepancies in the scientific literature about gut microbial phenotype in patients with endometriosis, there is no evidence of a decreased abundance of Faecalibacterium prausnitzii in patients with endometriosis, and this may suggest a lack of prebiotics deriving from the gut flora; it is the main product of SCFA among gut bacteria [30]. Anderson examined how inflammatory processes in endometriosis parallel an increased intestinal permeability in the context of dysbiosis [15]. Proinflammatory cytokines and pelvic pain upgrade intestinal permeability. Short Chain Fatty Acids (SCFA) play a key role in the gut immune barrier, regulating the pathogenesis of endometriosis. Indeed, butyrate is reduced in endometriosis [15]. Given the augmented intestinal permeability in endometriosis-affected patients, it could be interesting to examine whether leaky gut-related molecules—lipopolysaccharide (LPS) and zonula occludens-1 (ZO-1)—are increased in serum samples at different time points and during follow-up (FU) [31]. The GI tract hosts lymphoid structures and immune cells, affecting their formation and function, while gut flora also influences the mucosal composition of TH1, TH17, and Treg lymphocytes [32]. Indeed, in endometriosis, T-cell expression is imbalanced [33]. In women with stage 3/4 of endometriosis—according to the revised criteria of the American Society for Reproductive Medicine (ASRM)—TNFα was significantly increased [34]. Moreover, a greater decarboxylation of tryptophan favors the uptake of tyrosine. In the case of altered homeostasis, the uptake of tryptophan is reduced—and therefore, the brain synthesizes serotonin—generating stress [7]. Jiang et al. demonstrated that women with endometriosis often suffer from chronic stress—diagnosed by generalized anxiety disorder-7 and patient health questionnaire-9—which increases the growth of endometriotic lesions through activation of β-adrenergic signaling [4]. At the same time, in women with endometriosis, dysbiosis parallels chronic stress: on the genus level, an abundance of Paraprevotella, Odoribacter, Veillonella, and Ruminococcus is lower in chronically stressed endometriotic patients, identifying a microbial marker of disease. Furthermore, Immunohistochemistry highlighted an increased expression of proinflammatory cytokines—NF-κB p65 and COX-2—in chronically stressed patients with endometriosis [13]. Moreover, pre-clinical evidence found altered gut microbial composition in endometriotic mice with a relative abundance of Bacteroides. In addition, metronidazole acting on Bacteroides reduces inflammation and size of endometriotic lesions, suggesting that gut bacteria exacerbate endometriosis through inflammatory pathways [27,35]. Our study examines results from clinical trials and literature reviews and compares the gut microbial composition of patients with endometriosis in different contexts, but most data parallel gut microbiota to female reproductive tract (FRT) composition, losing focus on intestinal genera involved in the pathogenesis of endometriosis. Disease modulation may be provided by antibiotics, probiotics, and Fecal Microbiota Transplantation (FMT) [12]. Oral probiotic administration of Lactobacillus gasseri may relieve endometriosis-related symptoms and increase IL-12 concentration and reduce NK cell activity [36,37,38]. In the context of dysbiosis, a decrease in Ruminococcaceae is negatively correlated to apoptosis of epithelial cells and IL-6 levels in mice, enhancing peritoneal inflammation [39]. It was also found that letrozole restores the Firmicutes/Bacteroidetes ratio, α-diversity, and Ruminococcaceae abundance [40]. Regarding probiotics, it may be useful to administer patients with killed or inactivated bacterial taxa to cover the binding sites of endometriotic bacteria. Otherwise, compared to FRT microbiota, intestinal microbial composition related to endometriosis is far less investigated. Endometriosis has a multifactorial pathogenesis based on endocrine and immune disruptors, potentially associated with a core microbial profile, activating inflammatory pathways [12]. Increased proportions of Bacteroidetes in endometriosis and their reduced abundance after hysterectomy could prove the existence of a gut-estrogen axis [13,18]. In parallel, that could also be demonstrated by decreased proportions of Firmicutes in endometriosis and their higher abundance after hysterectomy [13,18]. In particular, according to the health-related quality of life (HRQoL) questionnaire, endometriosis-affected patients undergoing abdominal hysterectomy reported a significant improvement in QoL and endometriosis-related pain [7]. A clinical trial by Sandström et al. in 2020 stated that pelvic or lower abdominal pain in 137 women with endometriosis was significantly (28%, *p* < 0.001) and long-lasting alleviated after hysterectomy, according to QoL Registers and FU surveys [41]. There is evidence that laparoscopic hysterectomy should be preferred to laparotomy in severe endometriosis-affected patients since it is linked to fewer complications [42]. Surely, total abdominal hysterectomy—in which the uterus and cervix are removed—is controversial due to the long-term rebound of endometriosis-related pain and damage to ovarian circulation supply, affecting hormonal secretion. Although, the mechanism underlying a bidirectional estrogen-gut interaction is partially unclear. [13,18,43,44]. Despite the lack of evidence concerning the field, we shed light on an unexplored aspect of the disease. It would be appropriate to increase the number of studies in order to perform a further investigation on target treatment options.

## 5. Conclusions

The identification of an endometriotic phenotype in gut microbiota would pave the way for therapeutic approaches with probiotics before surgery. Moreover, the detection of dysbiosis—in the context of endometriosis’ clinical manifestations—would provide the clinician with new management strategies and prevent the progression of the disease. The main limitation of our paper is the heterogeneity of data and populations of patients. In addition, this review consists of a partial view of endometriosis. It is neither possible to determine whether gut microbial signatures are promoting or progressive factors in the pathogenesis of endometriosis nor their accuracy in early diagnosis. The strength of our work is that it is the first study to expressly resume the core profiling of bacterial taxa in the context of endometriosis. Further evidence is needed to investigate actual outcomes in clinical practice. For example, follow-up studies would reveal a possible role of microbial taxa in predicting prognosis or evaluating disease progression. Probiotics and prebiotics could be adjuvant treatments to surgery as targeted therapy in patients with a defined microbial profile.

## Figures and Tables

**Figure 1 medicina-59-00347-f001:**
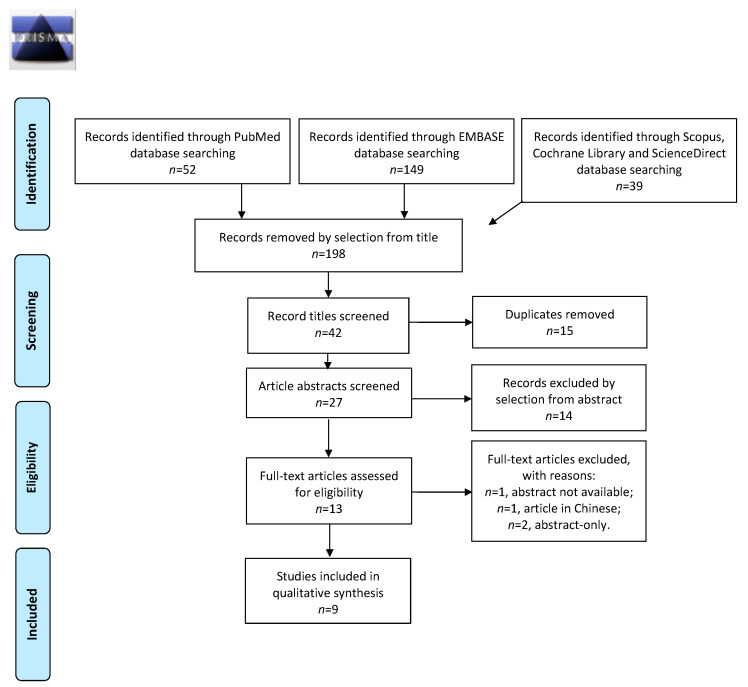
PRISMA 2009 Flow diagram showing records selection process.

**Table 1 medicina-59-00347-t001:** Characteristics of the studies.

Author, Year of Publication	Country	Study Design	No. of Patients	No. of Controls
[12]	Germany	NR		
[13]	China	CT	5	5
[14]	Turkey	CT	14	14
[15]	United Kingdom	NR		
[16]	Sweden	CT	66	198
[4]	Canada	NR		
[17]	China	CT	21	20
[18]	U.S.A.	CT	20	9
[19]	China	CT	12	12

CT: clinical trial; NR: narrative review.

**Table 2 medicina-59-00347-t002:** Outcomes of CTs at a taxonomic level in patients with endometriosis.

Author, Years of Publication	ASRM Stage	Treatment	Decreased Genera	Increased Genera	*p* Value
[13]	I-IV	N/A	Paraprevotella, Odoribacter, Veillonella, Ruminococcus,	Prevotella	*p* < 0.05
[14]	III-IV	N/A	Gardnerella, Snethia, Barnesella	Escherichia, Shigella	*p* < 0.01
[16]	N/A	Hormonal	Paraprevotella, Lachnospira, Turicibacter	Bacteroides, Parabacteroides, Oscillospira, Coprococcus	*p* < 0.05
[17]	I-IV	None	Clostridiales, Ruminococcus	Bacteroides, Blautia, Faecalibacterium prausnitzii, Bifidobacterium	*p* < 0.05
[18]	I-IV	Hormonal Surgery	Finegoldia, 1–68, Dialister, Lactobacillus	Faecalibacterium, Campylobacter	*p* < 0.05
[19]	III-IV	None	Lachnospira	Bifidobacterium, Blautia, Dorea, Streptococcus, Prevotella	*p* < 0.05

CT: clinical trial; ASRM: American Society of Reproductive Medicine.

## Data Availability

Data supporting the findings of the present study are available at reference numbers: [4,12,13,14,15,16,17,18,19].

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
