# Peer review of "Correlations between Gut Microbial Composition, Pathophysiological and Surgical Aspects in Endometriosis: A Review of the Literature"

_medicina, 2023, doi:10.3390/medicina59020347_

Round 1

Reviewer 1 Report

The aim of the paper “Correlations between Gut Microbial Composition,

Pathophysiological and Surgical Aspects in Endometriosis: a

Review of the Literature” is a review of the literature (6 clinical trials and 4 reviews) regarding the gut microbial composition in endometriosis and abdominal hysterectomy. The conclusion is that further studies are necessary for changing the therapeutic strategies.

Broad comments:

The manuscript has a modern concept, is well organized, adding value to the previous research concerning endometriosis and/or gut microbiota. The references comprise 44 articles. The findings are illustrated in 2 tables and 1 figure. Unfortunately, there are only 6 original studies regarding this feature, so the review should be extended to more relevant studies.

Specific comments:

The sections and subsections of the manuscript are appropriate.

In 1. Introduction section, the authors have presented information regarding endometriosis and dysbiosis. It is recommended to add some information regarding endometriosis characteristics, the aim of the study, and estimated results.

In 2. Material and Methods, the authors have described all the process of articles’ selection and statistical analysis. However, the limitation of the manuscript is evident, as there are only 6 studies regarding the gut microbiota in endometriosis, along with 4 reviews. It is strongly recommended to extend the authors search to have enough studies for a review, by changing your search words.

In 3. Results, the authors have described their findings in 3 subsections. It is strongly recommended to provide data regarding the clinicopathological characteristics of the patients, including endometriosis type, along with their therapies. Moreover, subsection 3.3 Gut microbiota of patients with undergoing abdominal hysterectomy is insufficiently presented. Information regarding the patients’ diagnoses prior the surgery is necessary and the relevance of association between abdominal hysterectomy and endometriosis gut microbiota.

In 4. Discussion section the findings are compared with other literature information. It would be useful to add more information regarding the importance of gut microbiota in the current therapy approaches of endometriosis. 

Furthermore, the section 5. Conclusions highlights the confirmation of their limitation, regarding the heterogeneity of data and populations. However, it is recommended to clearly state the value of the current study by comparison with other reviews.  

Author Response

Dear Reviewer,

Thank You for taking the time to review our manuscript and for your comments. They are crucial and valuable to us in raising the quality standard of our work.

We wanted to inform You that we have made a general revision of the English and grammar. In addition, a specification for Your revisions is below:

In 1. Introduction section, the authors have presented information regarding endometriosis and dysbiosis. It is recommended to add some information regarding endometriosis characteristics, the aim of the study, and estimated results.”

-Lines 40,41,45-49,63-67: We followed your advice and changed the introduction chapter according to the present observation.

In 2. Material and Methods, the authors have described all the process of articles’ selection and statistical analysis. However, the limitation of the manuscript is evident, as there are only 6 studies regarding the gut microbiota in endometriosis, along with 4 reviews. It is strongly recommended to extend the authors search to have enough studies for a review, by changing your search words.”

Thank You for your observation. We are please to explain that research was made through MeSH Terms. We specifically chose the term “Microbiota” (referring to a community of microorganisms) instead of “Microbiome” (referring to microorganisms’ genomic potential). That evidence in the context of endometriosis is extremely rare and poor.

In 3. Results, the authors have described their findings in 3 subsections. It is strongly recommended to provide data regarding the clinicopathological characteristics of the patients, including endometriosis type, along with their therapies. Moreover, subsection 3.3 Gut microbiota of patients with undergoing abdominal hysterectomy is insufficiently presented. Information regarding the patients’ diagnoses prior the surgery is necessary and the relevance of association between abdominal hysterectomy and endometriosis gut microbiota.”

-Lines 125-126: Thank You for your advice. We improved Table 2 with data regarding endometriosis stage and treatment. Subsection 3.3 is crucial for the Journal Special Issue, and it provides a general view of the influence of laparotomy on gut microbial composition. Unfortunately, there is a lack of data regarding microbiota, disease progression and follow-up in patients treated with hysterectomy, as we also specified in the manuscript.

In 4. Discussion section the findings are compared with other literature information. It would be useful to add more information regarding the importance of gut microbiota in the current therapy approaches of endometriosis.”

-Lines 366-393: We appreciated your comment and improved the discussion section in its respect.

Furthermore, the section 5. Conclusions highlights the confirmation of their limitation, regarding the heterogeneity of data and populations. However, it is recommended to clearly state the value of the current study by comparison with other reviews.”

-Lines 402-404: Thank you for your advice. We improved the conclusions section with the strength of our study.

Also, you can find the rewritten and corrected version of the manuscript in the attached file. We highlighted any changes made.

Thank You very much for your advice and comments. We hope we have complied with your requests.

Reviewer 2 Report

Review report

The manuscript entitled “Correlations between Gut Microbial Composition, Pathophysiological and Surgical Aspects in Endometriosis: a Review of the Literature” has prepared in proper manner and in my opinion it is qualified to be published in Medicina. However, some concerns were raised during the review process and need to be addressed to improve the quality of the manuscript.

Critiques:

1-      The manuscript submitted to be assessed is better to be line-numbered to ease pointing the issues. This for future submissions.

2-      Page 1 (P1): correct; the aim of this review is to, instead of was to.

3-      Resolution of figure-1 image is bad and blurry. Please improve the quality.

4-      During the screening method, what did authors mean by “duplicates were removed”? Please clearly explain.

5-      It was mentioned that 10 studies were chosen for this review article, however, table 1 shows characteristics of 9 articles only. Why?

6-      It is important and mandated to declare the population size of healthy controls and patients for each study mentioned in the table-2.

7-      Table-2: Shan et al reference showed that the dysbiosis was insignificant (P>0.05), this is incorrect. Also, in the original article of Shan et al, Prevotella was the only genus mentioned in the abstract that was significantly increased, while the other genera mentioned in the table 2 were mentioned as “significant differences” with no declarations whether they were increased or decreased. It is important in this article to describe the details clearly in your tables.

8-      Title of table-2 should be corrected to taxa instead of using the genera or genus, as the table is including different taxonomical levels.

9-      Table-2 should declare how the comparison was? Was it control vs patients or oppositely compared.

10-   Title of section 3.1 must be changed as mentioned in above point-8.

11-   P7: What does IBS mean? Please make sure that every acronym should be fully described at first time used.

12-   The introduction section lacks some more data about the role of gut microbiota in different aspects. In addition, the authors were focusing on tryptophan pathways and its dynamics throughout the microbiome as if the article is concentrated on this amino acid only. Including other amino acids effects could help to improve the introduction quality and give wider view to the reader.

13-   Conclusions could be improved if the authors simply correlate the findings of microbiomes with the prognosis, treatment and/or pathogenesis of endometriosis.

Author Response

Dear Reviewer,

Thank You for taking the time to review our manuscript and for your comments. They are crucial and valuable to us in raising the quality standard of our work.

We wanted to inform You that we have made a general revision of the English and grammar. In addition, a specification for Your revisions is below:

Page 1 (P1): correct; the aim of this review is to, instead of was to.”

-Line 18: We followed your advice and changed it according to your observation.

Resolution of figure-1 image is bad and blurry. Please improve the quality.”

-Lines 97,98: We found Your observation factual, so we have edited the image according to its respect and provided a better-quality figure.

During the screening method, what did authors mean by “duplicates were removed”? Please clearly explain.”

-Lines 87,88: Thank You for your observation. We followed your comment and edited the manuscript according to your advice.

It was mentioned that 10 studies were chosen for this review article, however, table 1 shows characteristics of 9 articles only. Why?”

-Lines 97,98,111,112,113,120,121: Thank You for your brilliant observation. We provided the incorrect number of records, due to a transcriptional error. We edited the text with the appropriate number of articles.

It is important and mandated to declare the population size of healthy controls and patients for each study mentioned in the table-2.”

-Lines 125-126: We followed your advice and changed it according to your comment.

Table-2: Shan et al reference showed that the dysbiosis was insignificant (P>0.05), this is incorrect. Also, in the original article of Shan et al, Prevotella was the only genus mentioned in the abstract that was significantly increased, while the other genera mentioned in the table 2 were mentioned as “significant differences” with no declarations whether they were increased or decreased. It is important in this article to describe the details clearly in your tables.”

-Lines 125-126: Thank You for your brilliant observation. We provided further details about our data.

Title of table-2 should be corrected to taxa instead of using the genera or genus, as the table is including different taxonomical levels.”

-Line 117: We found your comment factual, so we edited the table title in its respect.

Table-2 should declare how the comparison was? Was it control vs patients or oppositely compared.”

-Line 117: We found your observation factual, so we modified the manuscript according to your advice.

Title of section 3.1 must be changed as mentioned in above point-8.”

-Lines 129,188: We found your comment factual, so we edited the section title in its respect.

P7: What does IBS mean? Please make sure that every acronym should be fully described at first time used.”

-Line 299: Thank You for your advice. We edited the manuscript explaining the meaning of IBS.

The introduction section lacks some more data about the role of gut microbiota in different aspects. In addition, the authors were focusing on tryptophan pathways and its dynamics throughout the microbiome as if the article is concentrated on this amino acid only. Including other amino acids effects could help to improve the introduction quality and give wider view to the reader.”

-Lines 52-55,63-67: Thank You for your observation. We provided the introduction section with further details about microbial activity and we have highlighted them in the text.

Conclusions could be improved if the authors simply correlate the findings of microbiomes with the prognosis, treatment and/or pathogenesis of endometriosis.”

-Lines 400-408: We followed your advice and improved the conclusions section with more considerations about the role of microbiota in the pathogenesis of endometriosis.

Also, you can find the rewritten and corrected version of the manuscript in the attached file. We highlighted any changes made.

Thank You very much for your advice and comments. We hope we have complied with your requests.

Reviewer 3 Report

Authors present a systemic review based on the PRISMA recommendations devoted to the importance of gut microbiome changes for pathophysiology of endometriosis. The topic is not new, as several papers were published recently on this topic. I have some remarks concerning the manuscript:

1. The introduction should be rewritten in order to present the data about the microbiome changes both in genital tract and the gut in endometriotic patients, together with the explanation why the gut microbiome seems to be more relevant for the origin of endometriosis

2. Why have authors included a review repots in their analysis? They do not contain original results for interpretation?

3. The subchapter devoted to hysterectomy does not contain data being the main topic of the article. In my opinion should be removed. The discussion chapter is written incomprehensibly. The main points of the discussion should be presented as follows: the summary of the cited papers, the comparison with similar animal studies, the probable mechanisms of pathological connection between gut microbiome and endometriosis, namely changes in estrobolome, metabolome, immunosurveillance, cytokine secretion etc.

4. Conclusion should be more broadly written especially in the context of the possible application of gut microbiome composition in the diagnosis and treatment of the endometriosis. Authors should also propose some future directions in this field of investigation.

Author Response

Dear Reviewer,

Thank You for taking the time to review our manuscript and for your comments. They are crucial and valuable to us in raising the quality standard of our work.

We wanted to inform You that we have made a general revision of the English and grammar. In addition, a specification for Your revisions is below:

1. The introduction should be rewritten in order to present the data about the microbiome changes both in genital tract and the gut in endometriotic patients, together with the explanation why the gut microbiome seems to be more relevant for the origin of endometriosis”

-Lines 40,41,45-49: We followed your advice and changed the introduction section according to your observation.

2. Why have authors included a review repots in their analysis? They do not contain original results for interpretation?”

-Lines 125-126: Thank You for your observation. We are pleased to explain that we necessarily excluded review articles from our outcomes report because they did not contain results for interpretation. However, we considered including them in the records list since they belong to little evidence in scientific literature devoted to association between intestinal microbiota and endometriosis.Although, we improved Table 2 with data regarding endometriosis stage and treatment.

3. The subchapter devoted to hysterectomy does not contain data being the main topic of the article. In my opinion should be removed. The discussion chapter is written incomprehensibly. The main points of the discussion should be presented as follows: the summary of the cited papers, the comparison with similar animal studies, the probable mechanisms of pathological connection between gut microbiome and endometriosis, namely changes in estrobolome, metabolome, immunosurveillance, cytokine secretion etc.”

-Lines 269-293: Thank You for your brilliant observations. We assert that the subchapter concerning hysterectomy may be a hint for future perspectives in treatment and follow-up of endometriosis. Regarding the discussion section, we changed it according to your consideration.

4. Conclusion should be more broadly written especially in the context of the possible application of gut microbiome composition in the diagnosis and treatment of the endometriosis. Authors should also propose some future directions in this field of investigation.”

-Lines 400-408: We followed your advice and changed the conclusions chapter in its respect.

Also, you can find the rewritten and corrected version of the manuscript in the attached file. We highlighted any changes made.

Thank You very much for your advice and comments. We hope we have complied with your requests.

Round 2

Reviewer 1 Report

The aim of the paper “Correlations between Gut Microbial Composition,

Pathophysiological and Surgical Aspects in Endometriosis: a

Review of the Literature” is a review of the literature (6 clinical trials and now 3 reviews instead of 4 in the previous manuscript) regarding the gut microbial composition in endometriosis and abdominal hysterectomy. the cross-sectional investigation of a new method of diagnosis of tuberculosis infection in an endemic population. The conclusion is that further studies are necessary for changing the therapeutic strategies.

Broad comments:

The manuscript has a modern concept, is well organized, adding value to the previous research concerning endometriosis and/or gut microbiota. The references still comprise 44 articles. The findings are still illustrated in 2 tables and 1 figure.

Specific comments:

The sections and subsections of the manuscript are appropriate.

In 1. Introduction section, the authors have presented information regarding endometriosis and dysbiosis. There is some new information added, according the recommendations, regarding endometriosis characteristics, the aim of the study, but not of the estimated results.

In 2. Material and Methods, the authors have described all the process of articles’ selection and statistical analysis. However, the limitation of the manuscript is evident, as there are still only 6 studies regarding the gut microbiota in endometriosis. Although it was recommended to extend the search to have enough studies for a review, by changing the search words, the number of articles has been diminished by one (now 9 articles instead of 10).

In 3. Results, the authors have described their findings in 3 subsections. Although it has been strongly recommended to provide data regarding the clinicopathological characteristics of the patients, including endometriosis type, along with their therapies, only the stage and “hormonal” ± surgery has been added. Although it has been recommended to add information to subsection 3.3 Gut microbiota of patients with undergoing abdominal hysterectomy, only a last phrase has been added, without any information regarding the patients’ diagnoses prior the surgery and the relevance of abdominal hysterectomy and endometriosis gut microbiota.

In 4. Discussion section the findings are compared with other literature information. Some information regarding the importance of gut microbiota in the current therapy approaches of endometriosis has been added, without any mention about the huge limitations of the study, which is still practically based on 6 studies.

Furthermore, the section 5. Conclusions highlights the confirmation of their limitation, regarding the heterogeneity of data and populations. The value of the current study by comparison with other reviews is now mentioned, in association with exact bacterial taxa and possible future exploitation in therapy. 

Author Response

Dear Reviewer (1),

Thank You for taking the time to review our manuscript and for your comments. They are crucial and valuable to us in raising the quality standard of our work.

A specification for Your revision is below:

-We are aware of the limits of our study, which are mainly related to the lack of evidence about the field. We brought attention to an undiscovered issue in scientific literature, considering available data. Also in the Introduction and Discussion sections, we have highlighted that results of our study are strongly influenced by the exiguous number of existing works. However, we improved the Introduction and Discussion sections in order to highlight that aspect. 

Also, you can find the rewritten and corrected version of the manuscript in the attached file. We highlighted any changes made.

Thank You very much for your advice and comments.

Reviewer 3 Report

"The data about the microbiome changes both in genital tract and the gut in endometriotic patients, together with the explanation why the gut microbiome seems to be more relevant for the origin of endometriosis" - the introduction still presents the scarce information in this topic.

We assert that the subchapter concerning hysterectomy may be a hint for future perspectives in treatment and follow-up of endometriosis - fine, but the subchapter still lacks a clear connection between hysterectomy and endometriosis. It should be more emphasized.

Author Response

Dear Reviewer (3),

Thank You for taking the time to review our manuscript and for your comments. They are crucial and valuable to us in raising the quality standard of our work.

We are aware of the limits of our study, which are mainly related to the lack of evidence about the field. However, we shed light on an unexplored issue in scientific literature, using available data. Also in the Discussion section, we have highlighted that the expected results of our study are strongly conditioned by the small number of existing works.

Also, you can find the rewritten and corrected version of the manuscript in the attached file. We highlighted any changes made.

Thank You very much for your advice and comments.